# "People Who Fill the Spaces": Jodi Picoult and the Sarah Josepha Hale Award

**Jordan Hansen**

English Department, Literature and Criticirm, Indiana University of Pennsylvania, Indiana, PA 15701, USA; lrvcc@iup.edu or jordanhansen2012@yahoo.com

**Abstract:** This paper discusses the implications of Sarah Josepha Hale's polarizing opposition to the franchise of women on her legacy award, given to many authors since its inception, but most notably to contemporary women writers whom Hale likely would have rejected. In 2019, the award went to Jodi Picoult, an author who bridges journalistic writing on topics such as abortion, white supremacy, and gun violence, among others, with fiction novel writing. Hale's own works are archived through the Richards Free Library, and as such, the award is given for the entire collective body of work of one nominated literary person. The award impacts not only Picoult's career but Hale's legacy as an open opposer to the franchise of women, as well as the opportunities for contemporary women writers.

**Keywords:** Jodi Picoult; Sarah Josepha Hale; women writers; literary awards; feminist studies; realistic fiction; human rights

## 1. Introduction

Every year since 1956, the Richards Free Library in Newport, New Hampshire, has presented a literary award to some of New England's best writers "in recognition of a distinguished body of work in the field of literature and letters".[1] The award is named for Sarah Josepha Hale, an American writer, editor, and activist from New Hampshire, whose contributions to writing and American history have been lauded as essential to the history of women's writing and modern editorial practices. The award itself has gone to many notable New England writers including Robert Frost (1956) and Arthur Miller (1990), and in 2019, it was awarded to New York-born author Jodi Picoult.[2] According to the Richards Free Library website, the award cannot be applied for by the author or their publisher. Instead, the winners must be nominated by a Board of Judges who then vote based on five essential criteria:

- Nominees must have been born in New England, or reside there for at least part of the year as a regular practice.
- Nominees should be a literary person (poet, dramatist, novelist, historian, journalist, writer, etc.).
- Nominees must, if he/she doesn't meet the conditions of birth and residence, be associated primarily with New England through his/her work.
- The award is based on the full body of the nominee's work.
- Nominees must be able and willing to be present at the award ceremony and deliver a talk or reading of twenty to forty minutes duration.[3]

Winners are awarded a bronze medal with the likeness of Hale stamped in the center and the words "The Sarah Josepha Hale Award" circling her portrait, as well as a USD 2500 honorarium provided by the Holden-Yeomans Memorial Fund. More than that, the author is able to carry on Hale's legacy in conjunction with their own as a notable literary artist.

Hale's work as an author and editor during the late Enlightenment through the Antebellum period led her to produce a wide range of texts, with a particular focus on the women's sphere as it was shaping and re-shaping at the time of her work. Similarly,

Picoult's works over the last thirty years have gained notoriety for their social and political explorations, often landing her on banned books lists across the country for their arguably controversial nature. While both women have some qualities in common, most importantly their desire for bipartisan unification, they otherwise represent two very different kinds of activists. Part of this is due to the roughly 200-year separation in their writing periods, both women being products of their time. Since their careers are interconnected thanks to the literary award, it is those differences in their legacies that need to be addressed in order to understand how the legacies of women writers, past, present, and future, are inherently intertwined. Picoult's legacy is directly connected to Hale's due to the award, though Picoult's career is significantly more progressive and engaging than Hale might have preferred for a woman writer carrying on her name. Through an exploration of Hale's authorial and editorial history, as well as Picoult's more recent literary works, vocal activism, and collected interview responses, we can begin to piece together a larger picture of contemporary women's writing that is still evolving from the opportunities laid out by their predecessors.

## 2. A Transformation through Time: Sarah J. Hale's Authorial Evolution

Born in 1788 in Newport, New Hampshire, Sarah Josepha Hale née Buell grew up during the latter part of the Enlightenment period (late 17th century until the early 19th century) in a household that emphasized equal education for all of their children, regardless of sex or gender (Norwood 2017). As such, Hale was educated far beyond the typical level for women, affording her opportunities in education and authorial work that she otherwise may not have had available. The transition from the Enlightenment to the Victorian era brought on a drastic switch in ideologies surrounding women's education and Hale's own lines of thinking adjusted with the times. In her biography of Hale, Patricia Okker states:

> As Nina Baym has explained, postrevolutionary Enlightenment ideologies assumed that the mind has "no sex" so as to assert women's intellectual equality with men; in contrast, Victorian notions of "woman" constructed her as essentially different from man. This shift from equality to difference is key to Hale's own development as an editor. Though she would in some ways remain loyal to Enlightenment philosophies, during the late 1820s and the 1830s she gradually came to promote an essential sexual difference based on Victorian notions of women's inherent morality and the idea of a separate women's culture. These Victorian ideologies of gender, then, became the basis for her editorial career. (Okker 1995, pp. 38–39)

Therefore, while Hale was born and educated during the Enlightenment, she hit her stride as a writer during the Victorian period, writing Victorian ideologies for and about women using the methods she learned during her Enlightenment education. At best, Hale spent her career pushing against the very educational equality that she was benefitting from in the process.

Hale's education was, indeed, quite extensive despite the lack of opportunities for women at the time to attend formal schooling. However, her older brother Horatio, an attendee at Dartmouth College whom Hale described as 'very unwilling that I should be deprived of all his collegiate advantages,' compensated for holes in her education by tutoring her himself while he was home during school breaks (40). Thanks to her own thirst for knowledge combined with her brother's tutelage, Hale possessed the education needed to open a private school for local children when she was just eighteen, emphasizing a rather modern idea that children learn best, not by rote memorization and recitation but by learning to think critically and independently (41). Twenty years after she opened the school, Hale took over as editor of the *Ladies' Magazine*[4], during which time, Okker explains, "Hale started to shift toward Victorian notions of sexual difference soon after she turned to writing as an occupation. [ . . . ] During her editorship of the *Ladies' Magazine*, Hale vacillated between an Enlightenment emphasis on women's intellectual equality with men and a Victorian belief in women's moral difference from men" (44). It became apparent

quickly that Hale's avoidance of the complete acknowledgment of separatism was likely due to her wanting to avoid greater criticism. Unfortunately, though she started out with deeply seated Enlightenment ideals about there being no distinction between intellectual ability and gender, she quickly switched to a basic acknowledgment of the Victorian ideals regarding gendered difference, which she then facilitated often enough until she, too, began to actively endorse said ideals. Okker determines that "Hale's conversion to a Victorian perspective on gender was sealed in January 1837, when she began editing Louis Godey's *Lady's Book*, later titled *Godey's Lady's Book*" (50). It was during this time that Hale began to fully embrace separatist ideals regarding gendered abilities, namely in favor of the Victorian ideals that men are physical beings while women have a predilection for morality. This emphasis on less intellectually driven topics made up the majority of Godey's *Lady's Book* material, with "particular emphasis on subject matter that [Godey] felt was germane to the domestic realm of women such as fashion and dress, household management, the education of children, etc." [Sommers 2010, p. 44 (Godey 1850)]. No longer were women seen as exactly equal to their male counterparts but as separate entities with their own sphere of understanding on which they were intended to focus; namely, the domestic and maternal. As Okker goes on to explain:

> Hale's employment of this separatist and essentialist ideology, like that of many other women editors, was neither simple nor consistent. Even in the post-Civil War years, when American women's rights activists challenged notions of sexual difference in an effort to win political equality, Hale remained loyal to the separatist vision that she came to accept in the 1820s and 1830s. For Hale and for many other women editors, the metaphor of separate spheres remained an empowering rhetoric on which to base an editorial career. (58)

Much of Hale's work as the editor of this book reflected such ideals, with additional insistence on women's intentional separation from politics and public affairs outside the home.

As it turned out, however, Hale did not uphold the same separation of ideals in her writing and often used her editorial position to share her opinions with her readers, engaging in nationalist rhetoric disguised as harmless conversation starters and discussion board-style letters-to-the-editor within the pages of the *Lady's Book*. Against the general wishes of Louis Godey, who preferred to avoid such nationalist and political rhetoric, Hale "promoted domesticity and sentimentality with subversive political ends," and used the *Lady's Book* to create "opportunities for women to discuss issues that could not only enhance a national communal readership" and "advocated for both Northern and Southern women to juxtapose each other's otherwise unrelated lives" (Sommers 2010, p. 48). In essence, Hale was a strong advocate for women expressing their opinions on a myriad of political and nationalist topics, as long as they did so only in the privacy of their homes and remained distinctly bipartisan on any topic. It was this separation from the public aspect of such rhetoric that made her an advocate for women to stay out of politics altogether, and thus a vocal opponent of women's suffrage. Because her essentialist ideology still depended on separate spheres for men and women, women's suffrage undermined the idea that one sex was morally superior to the other, thus undermining her separatist work. While women's suffrage emphasized the idea that men and women were equal and should have equal representation through the voting process, Hale remained insistent that men and women were inherently different, and by giving women the right to vote, it would dismiss much of the progress made by women claiming their own world sphere and lead to partisanism.

Sommers seems to paint a rather negative opinion of Hale as "potentially little more than a very crafty editor, manipulating letters and opinion to broadcast her veiled polemic" (Sommers 2010, p. 51), while Okker insists that Hale's active efforts to appear bipartisan led Hale to lack opinions on certain topics altogether, including "whether women should become active in the debate about slavery" (Okker 1995, p. 82). However, this seemingly middle-ground stance did not transfer into her creative writing, particularly with her publication of *Liberia; or, Mr. Peyton's Experiments* (1853). This novel has been classified as

"plantation literature," a type of literary genre that emerged in response and opposition to anti-slavery and abolitionist literature of the Civil War era in order to depict slavery as beneficial to enslaved persons and less extreme than the supposed hyperbolic depictions in abolitionist literature such as Harriet Beecher Stowe's *Uncle Tom's Cabin*.[5] While Hale's book is not overtly pro-slavery, as other plantation literary works are, the plot involves a slave owner who wishes to free his slaves, but only if he can prove they would have a better life outside of forced servitude. After several attempts to free them, which all result in the freed people being bullied by white supremacists, Mr. Peyton determines that freed enslaved persons cannot prosper outside of slavery unless they are returned to their native land.[6] On the surface, this novel may seem to argue in favor of the freeing of enslaved persons, but it is clear that the message leans more toward perpetuating the otherness of African Americans in a free society by insisting that they have no place anywhere except in slavery or their native country. Considering the time in which this novel was written, it was likely viewed as uncharacteristically progressive, but it still manages to provide antebellum readers with a near-middle ground novel, allowing Hale to continue her efforts to remain bipartisan-adjacent and forming a legacy that is built on unspecified values regarding many otherwise pivotal topics.

*Liberia; or, Mr. Peyton's Experiments* is one of the only works of Hale's that is written strictly for adults, with much of the rest of her writing being either housekeeping guides, cookbooks, children's literature, and poetry, or what she considered "character sketches". These character sketches were compiled in *Traits of American Life* (1835)[7], *Sketches of American Character* (1838)[8], and *Northwood, or Life North and South: Showing the True Character of Both* (1852)[9], and consist of idealistic images of the lives of the "average" American. Given that these images are the works of Hale's ideas regarding what Americans could be like, the stories are most likely to be the best examples of what Hale felt were important traits in people and society in general. This is best evidenced in the preface to *Traits of American Life*, which specifically states that it was Hale's "intention, while displaying accurately, various traits in the American character, to furnish hints and examples which might be beneficial to society" (Hale 1835). While this article will not be going into detail about the character sketches within these books, we can, at least, speculate that a critical analysis of these sketches is likely to produce further detail into what Hale considered important characteristics of not only individual Americans but society as a whole, and provide more information about her personal beliefs socially and politically.

Considering Hale's authorial and editorial history, dedicating a literary award to her adds a significant amount of prestige to her name, and it might be speculated that much of her beliefs and/or history was intentionally overlooked in favor of focusing on her popularity in relation to the award's appeal. As we can recall, the award is given to an author from New England based on their entire body of work; therefore, Hale, who was from New England, was likely being judged on her entire body of work's commercial appeal rather than the individual aspects of it when choosing her for the award's namesake. That being said, if we are going to lay out the characteristics of her body of work against an author such as Jodi Picoult, who writes very different types of literary works to Hale, it is not enough to judge either of their works as a whole, but to look at the pieces that compile it.

### 3. Saying What You Mean: Jodi Picoult's Moral and Ethical Fiction

Jodi Picoult née Van Leer was born in Nesconset, New York, and has had an active writing career since 1992 with the publishing of her first novel *Songs of the Humpback Whale*. Since then, she has published 28 novels, five of which have been turned into movies, and currently has an off-Broadway musical which she co-wrote with her daughter Samantha prior to the pandemic which finally opened on the stage in June 2022. Along with the Sarah Josepha Hale Award (2019), Picoult has also won the New England Bookseller Award for Fiction, the Alex Award from the Young Adult Library Services Association (YALSA), a lifetime achievement award for mainstream fiction from the Romance Writers of America,

and the NH Literary Award for Outstanding Literary Merit and has two honorary Doctor of Letters degrees from Dartmouth (coincidentally, Hale's brother's alma mater) and from the University of New Haven.[10] However, it is the Hale award that appears the most interesting given the women's opposite approaches to writing.

Unlike Hale, Picoult takes a proactive stance in her writing regarding her position on an array of political and social topics and has not backed down from discussing those topics due to the possibility of losing readership. According to an April 2022 study completed by PEN America, Picoult is on the list of most frequently banned authors, with three of her books including *My Sister's Keeper* (2003) and *Nineteen Minutes* (2007) appearing on Banned Books lists across the country as recently as 2023.[11] Being on banned books lists may not seem like a raving review in regard to writing quality or style, but as Picoult joins other famously contentious authors such as Margaret Atwood, Toni Morison, and Angie Thomas on those lists, she, in turn, joins a long list of writers who are also unafraid to write the stories that need to be read, regardless of possible backlash. Prior to her winning the Hale Award, Picoult had produced a considerable list of books that explore a wide range of topics, including designer in vitro for medical purposes, school shootings, eugenics, and much more. Many of these novels have extensive critical analyses available, particularly *My Sister's Keeper*, *The Pact*, and *Nineteen Minutes* given their age, but there is little critical information yet available for anything newer than 2015.[12] For this article, however, I will put particular focus on three of her most recent novels as they exemplify the mix between journalistic exploration and narrative fiction that she has perfected during her career that earned her the award: *Small Great Things* (2016), *A Spark of Light* (2018), and *Wish You Were Here* (2021). Since she won the award in 2019, *Wish You Were Here* came after the win, but still deserves attention for the lengths to which Picoult went to create a timely and important novel.

Nearly one month prior to the pivotal 2016 presidential election, Picoult published *Small Great Things*, a novel written from multiple perspectives in order to provide the most well-rounded narrative experience given the deeply controversial themes. The story focuses on Ruth Jefferson, an African American labor and delivery nurse who has been ordered not to go near or make contact with the newborn baby of a white supremacist couple. When the newborn dies while in her care, Ruth is charged with murder, igniting a legal battle that raises questions about responsibility, ethics, and racial biases within the medical and legal systems. The story is told from multiple perspectives, which is characteristic of many of Picoult's novels. This one, in particular, is told from the perspective of Turk Bauer, father of the deceased child and a white supremacist, Kennedy McQuarrie, Ruth's public defender, and Ruth Jefferson. Each of them, as well as the other characters with whom they interact, experiences individual journeys of self-reflection based on racial prejudices, either as perpetrators or victims. The novel ends with each of the main characters going through an un-learning, of sorts, of the biases and prejudices they had been led to believe about people such as those they are facing off against (or with, in Ruth and Kennedy's case). The book received acclaim as an Indie Next pick and was a #1 New York Times best seller.

On her website, Picoult discusses in an extensive author's note her inspiration for this novel and the personal journey she went through in the process of writing it. At one point, she asks herself how writing about a person of color might be any different from writing about a white person, coming to the conclusion: "Because race is different. Racism is different. It's fraught, and it's hard to discuss, and so, as a result, we often don't" (Picoult 2016).[13] She discusses the real-life event that brought her the biggest bit of inspiration for her novel:

> Then I read a news story about an African-American nurse in Flint, MI. She had worked in labor and delivery for over twenty years, and then one day a baby's dad asked to see her supervisor. He requested that this nurse, and those who look like her, not touch his infant. He turned out to be a White Supremacist. The supervisor put the patient request in the file, and a bunch of African-American

> personnel sued for discrimination and won. But it got me thinking, and I began
> to weave a story. (Picoult 2016)

While this real-life story unfolded differently from the one Picoult outlines in her novel, the premise is the same. In the same author's note, Picoult also discusses her reasoning behind choosing such a dynamic and polarizing topic to write about, stating:

> Most of us think the word "racism" is synonymous with the word "prejudice".
> But racism is more than just discrimination based on skin color. It's also about
> who has institutional power. Just as racism creates disadvantages for people
> of color that make success harder to achieve, it also gives advantages to white
> people that make success easier to achieve. It's hard to see those advantages,
> much less own up to them. And that, I realized, was why I had to write this book.
> When it comes to social justice, the role of the white ally is not to be a savior or a
> fixer. Instead, the role of the ally is to find other white people and to talk to make
> them see that many of the benefits they've enjoyed in life are a direct result of the
> fact that someone else did not have the same benefits. (Picoult 2016)

It is clear, just from Picoult's own words about the work, why a novel such as this can be as important to the greater conversation about racism and prejudice in the United States as it appears to be. Clare Hayes-Brady has a brief explanation of the book's importance in her book chapter "Jodi Picoult: Good Grief," where she states that "Like her previous works, [*Small Great Things*] deftly engages with a thorny and divisive issue, and takes on complex current affairs with a light touch. However, the political and social immediacy of racial tension in the contemporary US may in future set this novel apart and see Picoult recognized as a more serious chronicler of her time" (Hayes-Brady 2018, p. 155). Taking Hayes-Brady's words into consideration, Picoult does appear to have created a novel that is self-reflective in regard to personal privileges. Instead, she has invited her readers to process the themes and events in the book in a way that helps them understand how they can be better activists (not just well-meaning allies) against injustices in their own lives and the lives of others.

Issues of race are far from the only thing Picoult has been willing to tackle in her novels. In 2018, *A Spark of Light* brought names and faces (however fictitiously) to another incredibly important topic that is still affecting people all over the country as recently as this year: abortion rights. In another multi-perspective piece, Picoult tells the story of a shooting at an abortion clinic in Mississippi where a gunman holds everyone inside the clinic hostage until his grievances against abortion services are heard. This book is organized through a series of timestamps, beginning at 5 p.m. on the day of the hostage situation, and counting backward each hour until 8 a.m. when it is revealed what happened just before each person found themselves at the clinic. There is also an epilogue at the end with a time stamp of 6 PM, which briefly details the aftermath of the hostage situation. Many voices are heard throughout the novel, including, but not limited to, Wren McElroy, who has come to the clinic with her aunt Bex, George Goddard, a distraught father who has decided the clinic is his ideal target to take his revenge on a system that he vehemently disagrees with, Hugh McElroy, the hostage negotiator and Wren's father, Janine Deguerre, an anti-abortion protestor who has entered the clinic undercover with a plan to discover exactly how sinister abortion clinics really are, and Beth Goddard, George's daughter who has recently obtained an illegal abortion that landed her in the hospital in critical but stable condition. Similar to *Small Great Things*, the characters experience a pivotal reevaluation of their preconceived notions about life and death, right and wrong, compassion and hatred, and must reconcile with themselves that what they may have felt was a non-negotiable stance is much more complicated than they previously considered.

Picoult includes an author's note at the end of this novel as well which includes the relevant historical implications that led her to want to write such an impactful story:

> Since 1977, there have been 17 attempted murders, 383 death threats, 153 instances
> of assault and battery, 13 individuals wounded, 100 stink bombs, 373 break-ins,

> 42 bombings, 173 arsons, 91 attempted bombings or arsons, 619 bomb threats, 1630 incidents of trespassing, 1264 incidents of vandalism, 655 anthrax threats, 3 kidnappings,[14] (Picoult 2018, p. 224)

As well as the legal and political battles surrounding women's reproductive rights that are still raging to this day. Through these interviews and research hours, Picoult works to repaint the pictures people have created in their minds about what it means to be a woman who seeks to have an abortion in the United States. As she explains in the author's note at the end of the novel: "Laws are black and white. The lives of women are a thousand shades of gray" (Picoult 2018, p. 226). This is particularly relevant now given the recent overturning of Roe v. Wade in July of 2022. In 2018 and now, women are still fighting for reproductive health and Picoult does not shy away from the need for people to consider how that will affect not only themselves but others around them.

It is important to mention, in regard to these two novels, the differences that can already be seen between Hale's writing and Picoult's writing. While both women come from a certain place of privilege within their own lives and social spheres, Hale's writing called for her readers to avoid confrontation, remain unbiased, and keep what opinions they do have within the home, out of sight. By contrast, Picoult calls for her readers specifically to take the themes and lessons from the story to heart and act on them. On her website page for *Small Great Things*, there is a section that starts with "I've read *Small Great Things*, and I want to DO something. HELP?!" Here, she makes clear that she knows she's not uniquely qualified to talk about race issues simply because she wrote a book about it, but does say, "I am not a social justice educator, so I can offer advice only as someone who is still a work in progress" (Picoult 2016). Picoult also includes suggestions on how to be a better ally and an active defender of marginalized groups including educating oneself about the lives of others, going outside of one's comfort zone in order to recognize their own privileges, and using one's platform to create discussions about issues that affect everyone, not just themselves. Essentially, Picoult calls for action while Hale called for quiet contemplation.

Picoult's most recent novel takes a different approach to journalistic storytelling in that the events discussed in it have little to do with social justice or societal rights. *Wish You Were Here* is set during the first wave of the COVID-19 pandemic and was published in 2021. There is no political agenda, ulterior motive, or underlying message about activism in this novel. It is a straightforward story set during the pandemic that captures the anxieties and turmoil that were experienced by everyone in the country when the COVID-19 lockdown first began. The novel tells the story of Diana O'Toole, an associate specialist at a major art auction company, and her experience with being stuck on an island in the Galapagos when the world shuts down to stop the spread of the virus. Her boyfriend, Finn, is a medical resident and is unable to accompany her on what was supposed to be a couple's trip. Stranded on an island that would otherwise be seen as a paradise, Diana deals with language barriers, restrictive living guidelines, and, thanks to the island's spotty internet connections, almost complete isolation from her life back in New York, including her partner. Finn does his best to keep her abreast of the developing pandemic situation at home, emailing her regularly with updates about the growing severity of the illness and the panic that hospitals are facing as they fight for control over a virus that is spreading faster than they can handle. The book begins in March 2020, right before the near-global lockdown is announced, and Finn's emails chronicle the events of the spread as they develop from mild worry to all-out panic. Considering the newness of this book in relation to the pandemic, readers find themselves re-living the events that led us to where we are now in 2023. For some, this might be similar to re-living a nightmare. For others, it might be cathartic to experience it from a distance, as a "remember when," even if that distance is not as distant as we could hope.

In the author's note at the end of the book, which is dated March 2021, exactly one year from the date the pandemic took hold, Picoult discusses how she felt as the virus spread, and how she chose to interview medical staff and COVID-19 survivors in order to get a deeper understanding of the way this virus affected people who experienced it in

ways she had not. She points out that, in her opinion, "The most pervasive emotion that we all have felt this past year is isolation. What's odd is that it's a shared experience, but we still feel alone and adrift" ([Picoult 2021](), p. 314). While her other two novels mentioned are stories of isolating experiences as well, this is something the collective can feel regardless of skin color or gender. She ends her author's note with the following self-reflective thought:

> When I try to make sense of the past year, it feels to me like the world pressed pause. When we stopped moving, we noticed that the ways we have chosen to validate ourselves are lists of items or experiences we need to have, goals that are monetary or mercenary. We don't need those things to feel whole. We need to wake up in the morning. We need our bodies to function. We need to enjoy a meal. We need a roof over our head. We need to surround ourselves with people we love. We need to take the wins in a much smaller way. And we need to remember this, even when we're no longer in a pandemic. ([Picoult 2021](), p. 317)

There is still the call-to-action that Picoult's novels tend to provide, to reflect and adjust on a personal level in order to move forward as a better person in spite of everything that otherwise keeps them down.

Some have been overtly critical of Picoult's works. Sarah Whitney calls her works "middlebrow" and asserts that the use of "multiple compelling narratives within the same novel suggests that the primary plots [ . . . ] are somehow insufficient themselves and must constantly be viewed through other lenses" ([Whitney 2016]()). By contrast, Hayes-Brady applauds Picoult's use of double plots and states that "One of the major attractions lies in its invitation to discuss socially relevant, often ethically thorny questions, and its provision of serious food for thought, thereby performing a very particular cultural task" ([Hayes-Brady 2018](), p. 148). I had the opportunity to ask Picoult herself (via email, which she responded to the same day she received it) about some of the critiques that have been made regarding her work, as well as how she would prefer to be received:

> **JH:** During my research so far, I came across a chapter in a book regarding your work. The author of the chapter, Clare Hayes-Brady, acknowledges the push against labeling works like yours and other women writers as "Chick-Lit," and instead likens it more to Sonya Andermahr's "women's grief fiction". Not that we need to put a label on things, but given the chance, how would you define your works instead? Do you think your works fall into a particular genre or category, or is there a greater variation of genres that might do it more justice? (I have an idea for this, but your opinion is the one I'm after).

> **JP:** I don't know that I write women's grief fiction, LOL. In fact, half my fan mail comes from men, who might be wholly surprised by that misnomer. I would define my novels as moral and ethical fiction. I certainly am not the first to write this—there's a long history, starting with Dickens and Austen. Who—of course—were also commercial writers, rather than literary ones.

> **JH:** Hayes-Brady points out the divide in critical and reader feedback in regard to the polemical topics discussed in your books. It's landed you on banned books lists across the country, but you still remain a decidedly popular author. How do you feel about this divide in reception, and what would you say to those who are wary about picking up your books (which are arguably people who need to the most)?

> **JP:** This is a difficult question. Part of what you're asking is what it's like to be branded a commercial fiction author when "clout" goes to literary authors. Commercial and literary are arbitrary marketing terms (people don't pick a book based on this designation). I will write the best book I can no matter what it's labeled, which means that I'd rather it reach more people—hence, my placement into commercial fiction marketing. That said, I know there are people who will not pick up a book I've written because it's not considered "highbrow" although I would absolutely argue that the quality of writing doesn't play into

> the terminology. The other half of your question is asking if books I write that have a particularly liberal slant (for example, ones about abortion rights, racism, trans issues) will alienate those who are living in conservative vacuums of social media and news. I don't know the answer to that. Some readers who have read me for ages will still pick up one of my books, because it's guaranteed to always give both points of view. My job, as I see it, isn't to get people to believe what I do. It's to lay out the facts and arguments on both sides, and ask you to ask yourself WHY your opinion is what it is.[15]

Picoult's determination to write books that argue both sides provides the opportunity to do exactly what she suggests: to ask one's self why they believe the way they do. If their opinion changes, then it might be considered progress. But regardless, the fact she does not shy away from presenting both sides rather than asking her readers not to consider either for the sake of nonpartisan standing is one of the many things that sets her apart from Hale.

## 4. Converging Legacies

Considering the methods of writing Hale chose, namely children's literature, editorials, and housekeeping guides, it can be stated that Hale also falls under the umbrella of commercial writing. Hale's reach during her time period alone was extensive enough to be considered popular and, since her death, it has become influential enough to dedicate a literary award in her honor. However, there is still the matter of her beliefs, which, as was mentioned previously, appear to have been overlooked in the decision to dedicate the award. Hale's work was partisan whether she advocated for it or not, and her book *Liberia; or, Mr. Peyton's Experiments* as well as her character sketches are evidence of that partisan thinking. Her use of the editorial as a way to advocate for nonpartisan thinking, and anti-suffragette ideals by connection, shows how far she was aware her words reached that she felt she could be a voice despite those conflicting messages.

On the other hand, Picoult uses her writing to explore difficult topics in order to inspire the kind of conversation that Hale expressed a desire to avoid. To clarify, Picoult does not tell everyone what they should be thinking, but rather that they should be thinking about it at all. Picoult clarified her feelings regarding her connection with Hale during the short interview:

> **JH:** Given Sarah Josepha Hale's background as a popular author but a noted anti-suffragette, how might winning an award tied to such a legacy tie into your own legacy as a passionate advocate for progress?
>
> **JP:** I have long been vocal about the misogyny inherent in publishing, and the fact that books written by women are dismissed as women's fiction. You don't see men being labeled as men's fiction, do you? You expect women to read widely, by authors of all genders, but men tend to read male authors. Why is that? What on earth is gendered about my writing? The answer? Nothing. And yet, by calling me a women's fiction author you're anticipating a subset of readers and excluding another set, which is frankly BS. We know that women are not reviewed as often as men, we know that they do not win as many literary prizes, and we know that they are often categorized as romance or women's fiction writers when that is an inaccurate label. I wouldn't say I'm doing the work of suffrage in my novels, but I would absolutely say that I am advocating for women to be heard, seen, and considered equal to any fiction being written by men today. In other words—let's just say that Sarah and I might agree to politely disagree:)[16]

Unfortunately, we will never know Hale's personal feelings on the matter, but as long as the award continues to go to authors whose entire body of work appears to run counter to Hale's ideals during her career, her legacy will continue to be tied to the progress and education of readers for which she originally advocated. This award builds a connection between two female writers working nearly 150 years apart, and while Picoult continues

to build her legacy on ideas of social justice, ethics, and morality, Hale's legacy is carried with her.

**Funding:** This research received no external funding.

**Institutional Review Board Statement:** Not applicable.

**Informed Consent Statement:** Informed consent was obtained by Jodi Picoult to use her interview in context.

**Data Availability Statement:** Not applicable.

**Conflicts of Interest:** The author declares no conflict of interest.

## Notes

1  Richard's Free Library, "Sarah Josepha Hale Award" (n.d.), Richards Free Library, https://richardsfreelib.org/about/sarah-josepha-hale/, accessed on 24 March 2022.
2  See complete list at https://richardsfreelib.org/about/sarah-josepha-hale/hale-award-winners/, accessed on 24 March 2022.
3  See note 2.
4  See "Sarah Josepha Hale | American author | Britannica". https://www.britannica.com/biography/Sarah-Josepha-Hale, accessed on 26 March 2022.
5  See "Plantation Tradition in Local Color Fiction" for more information about plantation literature https://public.wsu.edu/~campbelld/amlit/plant.htm, accessed on 10 May 2022.
6  Transcript of *Liberia; or, Mr. Peyton's Experiments* available at http://utc.iath.virginia.edu/proslav/halehp.html, accessed on 10 May 2022.
7  See pages on Google Books at https://books.google.com/books?id=mF4eAAAAMAAJ, accessed on 11 May 2022.
8  See pages on Google Books at https://books.google.com/books?id=n14eAAAAMAAJ, accessed on 11 May 2022.
9  See pages on Google Books at https://books.google.com/books?id=16oTAAAAYAAJ, accessed on 11 May 2022.
10 All of her achievements and a list of bestselling published books can be found in the bio on her website https://www.jodipicoult.com/JodiPicoult.html#code0slide1, accessed on 24 March 2022.
11 Complete list of frequently banned authors available on the PEN America website https://pen.org/banned-in-the-usa/#authors, accessed on 26 March 2022.
12 There is a Teacher's Guide for *Small Great Things* available through Penguin Random House found here: https://www.penguinrandomhouse.com/books/225537/small-great-things-by-jodi-picoult/9780345544971/teachers-guide/, accessed on 10 February 2023.; as well as a recent essay published by Artemis Michailidou: "Abortion and the Experience of US Citizenship in Jodi Picoult's A Spark of Light", *Women's Studies International Forum*, 92 (May–June 2022), 102578.
13 Entire author's note as well as an excerpt from the novel can be read on Picoult's website at https://www.jodipicoult.com/small-great-things.html#excerpt, accessed on 13 May 2022.
14 This is as of 2018 when the book was published. The numbers from 2022 have not yet been released, but the numbers for 2021 can be found on the National Abortion Federation website: https://prochoice.org/wp-content/uploads/2021_NAF_VD_Stats_Final.pdf, accessed on 17 Auguest 2022.
15 Email interview conducted on 20 May 2022.
16 From the same email interview.

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
