# Peer review of "“People Who Fill the Spaces”: Jodi Picoult and the Sarah Josepha Hale Award"

_humanities, doi:10.3390/h12020021_

Round 1
Reviewer 1 Report
This is a thoughtful reflection on the work and ideas of Jodi Picoult, taking as its focus her being awarded the Sarah Josepha Hale bronze medal. The award becomes a starting point for a consideration of Picoult and a comparison of her 21st century self-positioning with Hale's literary and social contexts in the early 19th century. It is clear in its aims and has a sensible structure. There really is very little criticism on Picoult and there's a bonus in the paper in the form of a brief interview with her.
Author Response
Dear Reviewer 1,
I greatly appreciate the time you took in reviewing and providing feedback on my manuscript. I am glad it was as concise and interesting as your comments made it seem, that was the goal.
Thank you kindly,
Jordan Hansen
Reviewer 2 Report
Lots of interesting material here, but you should clarify your own position in relation to the value of the Hale award, solidify your theoretical background and, most importantly, bring in your own energy! Good luck!

Reviewer 3 Report
There is a through-line from Hale to Picoult in American popular literature--middlebrow, commercial, or didactic. All of these are subsets of the middlebrow, which should have been examined more carefully, not just as chick lit or grief literature. Hale and Picoult serve similar functions in their immersion in politics, then moves away from it toward an aspirational objectivity (that neither seems to achieve). This is characteristic of the tradition of middlebrow writers, one that this article seems to call out for as context.
Round 2
Author Response
Dear Reviewer 2,
Thank you kindly for your second response to my manuscript and for taking the time to review it again. I appreciate your attention to detail. I am glad the objective os the manuscript is clear now.
Kindest regards,
The Author